# Individual Characteristics and Pain Sensitivity during Pregnancy—A Cross-Sectional Study in Pregnant and Non-Pregnant Women

**DOI:** 10.3390/ijerph192114151

**Published:** 2022-10-29

**Authors:** Katarzyna Leźnicka, Maciej Pawlak, Agata Gasiorowska, Aleksandra Jażdżewska, Dominika Wilczyńska, Paulina Godlewska, Anna Lubkowska, Monika Chudecka, Agnieszka Maciejewska-Skrendo, Rita Santos-Rocha, Anna Szumilewicz

**Affiliations:** 1Faculty of Physical Education, Gdansk University of Physical Education and Sport, K. Górskiego St. 1, 80-336 Gdansk, Poland; 2Department of Physiology and Biochemistry, Poznan University of Physical Education, Królowej Jadwigi St. 27/39, 61-871 Poznań, Poland; 3Faculty of Psychology in Wroclaw, SWPS University of Social Sciences and Humanities, Ostrowskiego 30b, 54-238 Wrocław, Poland; 4Department of Functional Diagnostics and Physical Medicine, Faculty of Health Sciences, Pomeranian Medical University, 54 Żołnierska St., 71-210 Szczecin, Poland; 5Institute of Physical Culture Sciences, University of Szczecin, 40B Piastów St., 71-065 Szczecin, Poland; 6ESDRM Sport Sciences School of Rio Maior, Polytechnic Institute of Santarém, 2040-413 Rio Maior, Portugal; 7CIPER Interdisciplinary Centre for the Study of Human Performance, Faculty of Human Kinetics, University of Lisbon, 1649-004 Lisbona, Portugal

**Keywords:** pregnancy, pain perception, optimism, personality trait, fear

## Abstract

The aim of the study was to describe the characteristics and factors related to pain perception in pregnant women, such as optimism, personality traits, and fear of developing COVID-19 consequences. Sixty-six pregnant women aged 23 to 42 years participated in the study, and the comparison group consisted of n = 59 non-pregnant female students aged 19 to 23 years. Pressure pain threshold and pain tolerance were measured with an algometer. To assess psychological characteristics, the Life-Orientation Test-Revised was used to assess optimism, the Fear of COVID-19 Scale was used to assess COVID-19 anxiety, and the Ten-Item Personality Inventory was applied to assess personality traits in a five-factor model. The main results of the study showed that pain tolerance was significantly lower in both dominant and non-dominant hand pregnant women than in the comparison group. The studied pregnant women had higher scores for conscientiousness, fear of COVID-19, and optimism compared with the non-pregnant women. Regression analysis showed that the variability in pain perception among pregnant women could not be explained by individual differences in personality traits, optimism, and fear of COVID-19.

## 1. Introduction

Pregnancy usually involves significant anatomical and physiological changes to meet the demands of the placenta and fetal development [1,2]. These changes affect the functions of each organ and require complex coordination by placental and non-placental hormones to ensure adequate transfer of nutrients from the mother to the fetus [3]. One of the most important maternal hormones that increase significantly during pregnancy is oxytocin [4], which regulates, among other things, the cardiovascular system of the pregnant woman and the fetus [5], enhances mood and well-being, reduces anxiety and pain decreasing physiological and psychological stress [6]. There is also evidence that oxytocin plays an essential role in the psychosomatic adjustment of the pregnant woman to the state of pregnancy and motherhood, as well as in the adaptation of the newborn to the new environment [6]. A similar role is attributed to the endorphin system, which is activated in late pregnancy and during childbirth. The high endorphin level not only has an analgesic effect but also influences the quality and strength of bonding with the baby [7]. However, pregnancy is associated with a series of changes in the psyche, lifestyle, and social roles that have the greatest impact on women’s daily lives. These processes are further modulated by anxiety, stress, and lack of social support, leading to an increased risk of depressive disorders in up to 23% of pregnant women [8].

An important factor influencing pregnant women’s emotions and behavior is pain, a subjective phenomenon that appears to play an important role in women’s attitudes toward pregnancy and childbirth. The significant pain experienced during childbirth can lead to pathological effects both antepartum and postpartum. Childbirth pain has a significant impact on the fear of childbirth (FOC), a common problem affecting women’s health and well-being during pregnancy and postpartum [9]. Fear of childbirth is a complex and multifaceted problem that can lead to complications in approximately 20% of pregnancies [10]. In general, high fear of childbirth (FOC) and anxiety in pregnant women is associated with low levels of emotional stability, introversion, low agreeableness, lack of experience with childbirth, and low self-esteem [11,12,13]. These characteristics may also manifest in certain limitations in establishing social relationships, which translates into lower levels of social support, especially in coping with birth anxiety [14,15]. In addition, the fears and anxieties experienced during pregnancy, as well as well-being during pregnancy and postpartum, may also be influenced by situational factors, including global events such as the pandemics of COVID-19. Restrictions implemented in the context of this disease to reduce infection rates led to social confinement. They caused pregnant women to suffer not only in their daily lives but also in their medical care experience during pregnancy. Many pregnant women feared COVID-19 and, therefore, shied away from visiting delivery rooms and maternity wards, postponing or even canceling prenatal visits and face-to-face counseling sessions during the pandemic [16,17,18,19]. Their anxiety was heightened by the thought that the virus could be transmitted to the fetus through vertical transmission [20]. As a result, the psychological well-being and mental state of many expectant mothers deteriorated significantly [21].

At the same time, psychological traits such as optimism might be associated with a higher birth quality. According to Scheier and Carver (1987), optimism as a personality trait means general optimistic expectations and a tendency toward positive feelings and life satisfaction. Many studies have shown that optimists respond more positively to health-threatening situations [22] and use task-oriented coping strategies, switching to more adaptive emotional strategies such as acceptance and humor when needed. They rely on their resources to overcome difficulties and adapt to change more easily than pessimists. Dispositional optimism is highly correlated with a sense of self-control and self-efficacy, which is also crucial for the birth experience [23,24]. Some other studies have also shown that personality is related to subjective pain perception, which was confirmed in Iranian women: those who exhibited higher extroversion were more likely to report severe pain during delivery. In this studied population, neuroticism and openness to experience were associated with subjective measures of labor pain, such as a stronger response to pain, higher pain intolerance, and greater depth of pain [25]. In addition, introverted and less agreeable women may have some limitations in establishing social relationships, which translates into lower levels of social support, particularly in coping with birth-related anxiety [14,15].

In sum, all these factors—as well as pregnancy-related symptoms—are not insignificant for women’s health-related quality of life [26]. Moreover, the perception of quality of life in pregnant women, especially concerning physical performance, is lower than in non-pregnant women of similar age [27]. It is worth noting that this component of quality of life decreases during pregnancy while the psychological component remains stable [28].

In conclusion, this study aimed to investigate the specificity and factors related to pain perception in pregnant women. Unlike most studies that examine subjective ratings of persistent pain, pain tolerance, or pain sensitivity with simple self-report measures, this study examined pain perception in response to applied pressure pain stimuli. It assessed these mechanical stimuli quantitatively and qualitatively. Given the subjective nature and broad spectrum of pain modulation, this study also examined psychological factors that may determine pregnant women’s psychological well-being and thus influence their pain experiences, such as optimism, personality traits, and developmental anxiety COVID-19. The group of pregnant women consisted of n = 66 voluntary participants enrolled in a supervised exercise and education program at the College of Physical Education and Sport in Gdansk. The comparison group consisted of physical education students whose physical activity levels may be higher than those of pregnant women. Since the level of physical activity could lead to increased pain tolerance [29,30], this potential confounder was controlled for in the study.

## 2. Materials and Methods

### 2.1. Participants

The study included *N* = 125 healthy women aged 23–42 (*M* = 26.41, *SD* = 6.04). The focus group consisted of *N* = 66 pregnant women (*M* = 31.27, *SD* = 4.17), participants of a supervised exercise and educational program aimed at pregnant women. On average, they were in the *M* = 21.95 weeks of normal pregnancy (*SD* = 4.44, range 13–28). Most of them had not previously given birth (n = 44, 66.7%). The comparison group consisted of *N* = 59 females, non-pregnant students from the Gdansk University of Physical Education and Sport in Poland, aged 19–23 (*M* = 20.97, *SD* = 1.20). The comparison group was significantly younger than the group of pregnant women, *t*(76.91) = −19.20, *p* < 0.001. Four participants from this group did not fill in the questionnaires. They were excluded from the analyses concerning psychological traits.

### 2.2. Pain Measurement

PPT, PTOL. Tissue pressure sensitivity measurements were performed using an algometer from Medoc AlgoMed (Israel). The AlgoMed system allows the user to perform various pressure testing protocols at selected points on the body. The device has a digital head that is connected to a computer. The researcher places the head at the test site and presses it on the test site with increasing speed (kPa/s). The result is visible in the head window and stored in the computer’s corresponding file.

Two tests were performed using this device to determine pain threshold (PPT) and pressure pain tolerance (PTOL). PPT is defined as the minimum amount of pressure required for the sensation of pressure to first change to pain, while PTOL describes the maximum stimulus intensity or duration of continuous painful stimulation that a subject is willing to endure.

Participants were instructed on how to use the algometer and then allowed to use the device. They were tested in a sitting position, and measurements were taken on both upper limbs on the dorsum of the hand between the thumb and index finger. All measurements were taken by the same researcher, and always in the morning. The researcher placed the algometer head on the area to be examined and gradually applied stimuli at a rate of 30 kPa/sec. When pain occurred, the participant said “stop”, and this measurement was used as an indicator of pain threshold (PPT result). The measurement was then continued until the participant could no longer tolerate the stimulus and signaled the end of the measurement. The point at which a painful pressure stimulus could no longer be tolerated was used as the pain tolerance measurement (PTOL result).

Visual Analog Scale (VAS). After completing the PPT and PTOL tests, patients were asked to report their subjective pain level using the Visual Analog Scale (VAS) to assess the extent of subjective pain during the procedure compared with an individual’s subjective pain experience. Pain intensity was rated using a scale from 0 = “no pain and discomfort” to 10 = “the worst possible pain and discomfort”.

### 2.3. Psychological Traits

Optimism. Optimism was measured using the Life-Orientation Test-Revised (LOT -R), a 10-item measure of optimism or pessimism developed by Scheier and Carver (1985) and translated into Polish by Poprawa and Juczynski [31]. Of the ten items, six measure optimism (e.g., In uncertain times, I usually expect the best, and If something can go wrong for me, it will, with reverse scaling), and four items serve as fillers. Respondents rated each item on a 5-point scale from 0 = strongly disagree to 4 = strongly agree. Responses to the six items were summarized so that a higher score indicated greater optimism (range 6–24, *M* = 16.35, *SD* = 3.79, α = 0.71).

Fear of COVID-19 (FCV-19S). Fear of not contracting COVID-19 was measured using the Fear of COVID-19 Scale [32,33]. The scale consists of 7 items (e.g., I am uncomfortable thinking about coronavirus-19; I am afraid of losing my life because of coronavirus-19), which were rated on a 5-point Likert scale ranging from 1 = strongly disagree to 5 = strongly agree. The sum of the responses indicated the level of fear of coronavirus COVID-19 (range 7–31, *M* = 12.00, *SD* = 4.17, α = 0.84).

Personality traits. We measured personality traits within the Five-Factor Model using the Polish version of the Ten-Item Personality Inventory (TIPI-PL); [34,35]. It consists of two items per each of five scales measuring extraversion (e.g., I see myself as extraverted, enthusiastic), agreeableness (e.g., I see myself as critical, quarrelsome; reverse scored), conscientiousness (e.g., I see myself as dependable, self-disciplined), emotional stability (e.g., I see myself as anxious, easily upset; reverse scored), and openness to experience (e.g., I see myself as open to new experiences, complex). Items are rated on a 7-point Likert scale from 1 = disagree strongly to 7 = agree strongly. The higher the specific score, the higher the level of the personality trait (range 2–14, extraversion *M* = 11.91, *SD* = 2.14; agreeableness *M* = 11.51, *SD* = 1.91; α = 0.36; conscientiousness *M* = 11.19, *SD* = 2.40, emotional stability *M* = 9.12, *SD* = 3.13; openness to experience *M* = 10.35, *SD* = 1.82; αs < 0.63).

### 2.4. Physical Activity

The level of physical activity in pregnant women was measured by the short form of the International Physical Activity Questionnaire [36,37]. This questionnaire, which has shown acceptable measurement properties, provides information on weekly physical activity levels in multiples of the resting metabolic rate (METs). Based on IPAQ outcomes, the pregnant women were categorized into three levels (categories) of PA: low (inactive women), moderate (accumulating a minimum recommended level of PA), and high (exceeding the minimum recommended level of physical activity) [38]. The level of physical activity in the comparison group was not measured. However, the students for the research were recruited in the later year of their studies, so we assumed they undertake a high level of physical activity due to compulsory sports activities connected with the curriculum.

### 2.5. Ethics Statement

The Bioethics Committee of the Regional Medical Chamber in Gdansk (Poland) approved the study (KB-8/21). The investigation protocols were conducted ethically according to the World Medical Association Declaration of Helsinki. The participants were informed about the purpose of the experiment and gave written informed consent to participate in the research. All personal information and results were anonymous and were processed and stored following current regulations of data protection in Poland. This study is a part of a clinical trial registered in ClinicalTrials.gov (NCT05009433).

### 2.6. Statistical Analysis

The analysis was conducted in JAMOVI [39]. The threshold for statistical significance was set at *p* < 0.05. Although the number of participants in the pregnant group and in the non-pregnant group did not differ (*p* = 0.592), the distributions of the pain measurement variables and all psychological variables significantly differed from a normal distribution (*p*s < 0.001 in the Shapiro–Wilk test, only for Optimism *p* = 0.034). Hence, the *t*-test with Welsh correction was used to compare two research groups, and the Wilcoxon rank test was used to compare two measurements within each group.

The sensitivity analysis using G*Power [40] revealed that with power 1-β = 80% and significance α = 0.05, a sample of this size is large enough to detect an effect of Cohen’s *d* = 0.51) in two-groups comparison, and an effect of R^2^ = 6% in linear regression with a single predictor.

## 3. Results

Table 1 presents the results of a comparison between pregnant and non-pregnant women in terms of pain measurement, while Table 2 presents the results of the corresponding comparison for psychological variables. The results show that pain tolerance in both the dominant and non-dominant hands was significantly lower in the pregnant women than in the comparison group. In turn, the scores for optimism, conscientiousness (personality trait), and fear of COVID-19 were higher in the pregnant women than in the non-pregnant group. No significant differences were found for the remaining variables.

In the second step, we compared pain sensitivity concerning all three indicators (pain threshold, pain tolerance, and subjective pain sensitivity) when measured on the dominant hand and the non-dominant hand, using the Wilcoxon rank test. We found that the data of pain threshold measured on the dominant hand was significantly lower compared with the non-dominant hand for the whole sample, *W* = 4931, *p* = 0.008, Cohen’s *d* = −0.277 and non-pregnant women, *W* = 1181, *p* = 0.025, Cohen’s *d* = −0.360, but not for pregnant women, *W* = 1292, *p* = 0.152, Cohen’s *d* = −0.191. In view of the pain tolerance, it was significantly higher when measured on the dominant hand than on the non-dominant hand for the whole sample, *W* = 2701, *p* = 0.002, Cohen’s *d* = 0.229, for non-pregnant women, *W* = 620, *p* = 0.046, Cohen’s *d* = 0.299, and for pregnant women, *W* = 759, *p* = 0.027, Cohen’s *d* = 0.263. Finally, subjective pain assessment was significantly lower when measured on the dominant hand than on the non-dominant hand, for the whole sample, *W* = 1552, *p* < 0.001, Cohen’s *d* = 0.473, and for pregnant women, *W* = 251, *p* < 0.001, Cohen’s *d* = 0.674, but not for non-pregnant women, *W* = 521, *p* = 0.075, Cohen’s *d* = 0.272.

Next, Spearman’s correlations between pain sensitivity (pain threshold, pain tolerance, and subjective pain sensation measured on the dominant and non-dominant hand) and psychological variables were analyzed in the total sample and separately for the pregnant women and the comparison group (see Table 3). This analysis revealed negative correlations between optimism and pain threshold, especially in pregnant women. In addition, pain tolerance correlated significantly and negatively with optimism and fear of COVID-19 and, to some extent, with conscientiousness in the entire sample but with agreeableness in the group of non-pregnant women. Correlations for subjective evaluation of pain included a positive association with extraversion in the total sample and additional positive correlations with optimism and stability in the group of pregnant women. Importantly, the psychological traits that were correlated with indicators of pain sensitivity differed in the two groups (pregnant and non-pregnant). Therefore, a supplemental general linear regression analysis was performed to control for variables (optimism, personality traits, and fear of COVID-19) as additional covariates for the effects of pregnancy on pain sensitivity. In addition, physical activity level was also included as a potential confounder. All variables were z-scored before analysis to obtain standardized results. The results of these correlations are shown in Table 4.

The regression model for the pain thresholds in the dominant hand was marginally significant, *F*(10, 110) = 1.90, *p* = 0.053, η^2^_p_ = 0.147. A higher pain threshold was associated with older age and a lower level of optimism. Controlling for all additional variables, no significant differences between pregnant women and the comparison group concerning the pain threshold were found (Table 4). In turn, the model for the pain thresholds in the non-dominant hand was significant, *F*(10, 110) = 2.00, *p* = 0.040, η^2^_p_ = 0.154. Moreover, it was found that the higher pain threshold was also associated with older age, but also a lower level of conscientiousness and a higher level of agreeableness. Controlling for all additional variables, no significant differences between pregnant women and the comparison group concerning the pain threshold were found (Table 4).

The regression models for pain tolerance were significant both for the dominant hand, *F*(10, 110) = 6.99, *p* < 0.001, η^2^_p_ = 0.388, and non-dominant hand, *F*(10, 110) = 5.80, *p* < 0.001, η^2^_p_ = 0.345%. Controlling for all additional variables, we found that pain tolerance in pregnant women was significantly lower than in non-pregnant women (Table 4). Finally, the regression models for subjective pain assessment were insignificant both for the dominant hand, *F*(10, 110) = 0.158, *p* = 0.121, η^2^_p_ = 0.126, and the non-dominant hand, *F*(10, 110) = 1.53, *p* = 0.138, η^2^_p_ = 0.122. In both cases, higher subjective pain assessment was associated with higher extraversion and lower agreeableness. Controlling for the covariants, the differences between pregnant and non-pregnant women’s subjective pain sensitivity were insignificant (Table 4).

In addition, predictors of pain perception were analyzed only in the group of pregnant women. This analysis included all predictors tested in the previous analyses, as well as the week of pregnancy and the variable representing the number of previous deliveries. However, the overall model was not significant for pain threshold [dominant hand: *F*(11, 54) = 1.54, *p* = 0.144, η^2^_p_ = 0.239, non-dominant hand: *F*(11, 54) = 1.48, *p* = 0.167, η^2^_p_ = 0.231], pain tolerance [dominant hand: *F*(11, 54) = 0.98, *p* = 0.472, η^2^_p_ = 0.167, non-dominant hand: *F*(11, 54) = 0.79, *p* = 0.649, η^2^_p_ = 0.139], and subjective perception of pain in relation to the dominant hand: F(11, 54) = 1.67, p = 0.105, η^2^_p_ = 0.254. In turn, the regression model was significant for subjective pain perception in relation to the non-dominant hand: *F*(11, 54) = 2.61, *p* = 0.010, η^2^_p_ = 0.348. The level of the dependent variable was associated with high levels of extraversion (β = 0.37, se = 0.13, *t* = 2.77, *p* = 0.008) and was higher in women who had given birth in the past (β = 0.69, se = 0.27, *t* = 2.58, *p* = 0.013). Overall, the variability in pain sensitivity among pregnant women cannot be explained by individual differences in personality traits, optimism, and fear of COVID-19, as well as by participant age, gestational week, and the number of previous births.

## 4. Discussion

The sensation of pain is a highly individual and psychological experience, perceived peripherally but originating in the brain. This sensory phenomenon is modulated to a considerable extent by the circumstances in which it occurs, primarily environmental, social, emotional, or religious circumstances [41]. However, pregnancy is also a factor that undoubtedly affects the perception and processing of pain, which in turn influences the psychology and physiology of women. In this case, the spectrum of potentially modulating factors increases including anxiety, previous birth experiences, social environment, educational level, unemployment, and perinatal information events. The main objective of this study was to investigate the relationship between psychological characteristics such as personality traits, optimism, and fear of COVID-19 and pain perception in pregnant and non-pregnant women. Our results showed that the pregnant women had significantly lower pain tolerance but higher scores for optimism, conscientiousness, and fear of COVID-19 than non-pregnant women. No differences were found between the two groups in pain threshold, but only in view of pain tolerance. Due to the age differences between pregnant and non-pregnant women, we controlled for the age of the participants (Table 4). As there were no differences in pain threshold between the two groups studied, the issue of age differences seems to be irrelevant in this case. In addition, for pain tolerance, no significant effect of pregnancy and no effect of age was found, and for subjective pain sensitivity, were found no effect of pregnancy and age. The results obtained indicate that the differences in pain tolerance between the two groups were not caused by age or life stage. Regarding pain perception, this study is conceptually related to the results published by Goolkasian and Rimer [42]. They thermally stimulated pregnant and non-pregnant women nine times between the sixth month of pregnancy and the puerperium. At the same time, the study was conducted on non-pregnant women. Interestingly, the pain response did not change over time in the control women, whereas the pregnant women in the last two weeks of pregnancy reported that the applied thermal radiation was perceived as more painful than in the other stages of pregnancy. Studies by Cogan and Spinnato have shown that the pain threshold increases in the last 10 days of pregnancy for both noxious pressure stimuli and thermal radiation [43]. In contrast, in thirty pregnant women, Sengupta and Nielsen found no change in pain threshold during induced labor (artificial rupture of membranes only) [44]. Staikou et al. [45] also showed no differences in pain threshold in response to mechanical and electrical stimuli in pregnant and non-pregnant women. The results obtained in this study may indirectly indicate that an analgesic effect of released β-endorphins does not affect pain perception in late pregnancy, suggesting that the central pain suppression system is not active in late pregnancy. However, animal studies showed that the significantly higher estrogen and progesterone levels and activation of the endorphin system cause pregnancy-related antinociceptive effects [46]. Studies in pregnant women have yielded conflicting results. In several clinical studies, researchers have described an increase in heat pain tolerance (HPTo) during labor compared with a non-pregnant control group and postulated it as the cause of hormonal changes during pregnancy [47,48,49]. However, hormonal studies have not supported these results. In contrast, Frolich and colleagues confirmed the presence of differences in estradiol and progesterone levels between the term and postpartum visit; however, thermal pain tolerance did not change significantly [50]. These study results argue against the concept of simple progesterone- or estrogen-induced analgesia in humans and do not support the hypothesis that late pregnancy is associated with increased antinociception in women. In future studies, it may be valuable to assess the levels of β-endorphins, progesterone, and estrogen in study participants and correlate these biomarkers with the pain sensitivity in both pregnant and non-pregnant women. Another interesting hormone that should be tested in the context of pain management should be oxytocin. Thus far, the existence of biologically and psychologically plausible mechanisms linking oxytocin and pain have been well supported using animal models with limited but encouraging human research [51]. However, the data on the role of oxytocin in the management of pain during pregnancy are not sufficient to use for therapeutic purposes.

The current study results show that pregnant women have significantly higher levels of optimism than non-pregnant women. Individuals with high levels of optimism expect positive outcomes for their future and are better able to cope with stress and everyday challenges [22]. High levels of optimism have been associated with better coping with stressful situations and better physical and psychological well-being [52,53], so it is plausible that it is related to lower pain sensitivity. In our study, no association was found between optimism and pain sensitivity. However, pain tolerance was significantly and negatively correlated with optimism in the entire sample, just as higher levels of optimism were associated with less pain. It can be speculated that optimism is related to pain experience or pain sensitivity in two different ways: negatively, by reducing feelings of hopelessness, and positively, by increasing pain acceptance. These two mechanisms could then suppress each other, resulting in a nonsignificant relationship between optimism and pain. This hypothesis should be investigated in future studies examining the precise mechanisms of the relationship between optimism and pain sensitivity in pregnant and non-pregnant women.

Our results clearly show that the pregnant women studied had higher levels of conscientiousness and fear of COVID-19 compared with non-pregnant women. Conscientiousness was also the predictor of pain threshold in the non-dominant hand. These findings are consistent with those of Conrad and Stricker, according to which pregnant women with higher levels of conscientiousness reported more positive experiences during labor and delivery than pregnant women who rated this trait as low [54]. This could be due to the fact that more conscientious mothers are more likely to deal with all aspects of labor and delivery and feel more confident afterward, which could lead to more pleasant feelings. In addition, better preparation for birth could help to avoid surprises during the birth process and to use task-oriented coping strategies when needed. In addition, the present study confirmed the negative associations between pain tolerance and agreeableness in non-pregnant women and the positive correlations between extraversion in the overall sample and emotional stability in pregnant women. Individuals high in trait agreeableness were able to cope with pain while internalizing it, maintaining negative emotions, and experiencing more pain. On the other hand, extroverted and emotionally stable pregnant women are likely to be better at expressing their emotions and thus better able to cope with pain. They face challenging situations with greater composure and rationality and use more task-oriented strategies to cope with pain and other difficulties [55].

The significantly higher level of fear of COVID-19 in pregnant women compared with non-pregnant women is likely the result of the health and economic complications affecting pregnant women during the pandemic outbreak, as well as uncertainty about the effects of COVID-19 on the fetus. Research shows that pregnant women fear both continuing the pregnancy and endangering their own lives and the need to terminate the pregnancy as a result of infection because they believe they are more likely to develop a severe course of infection and that the infection could be transmitted to their unborn child [56]. Negative feelings and emotions can influence pain perception and increase pain perception [57]. This was also confirmed by the results of our study related to anxiety, where pain tolerance was significantly and negatively correlated with fear of COVID-19 in the whole sample. Negative thoughts influence subsequent fear of childbirth during pregnancy, which could lead to increased pain perception. However, in our study, the significantly higher COVID-19 anxiety scores in pregnant women compared with non-pregnant women were not associated with pain sensitivity scores when controlling for personality traits and optimism. This could be because the pregnant women in this study had simultaneously higher optimism and COVID-19 anxiety scores than the nonpregnant women and that these two factors cancel each other out. The authors are aware that the present study is not free of limitations. First, although our sample seemed large enough to warrant sufficient statistical power, it is possible that the effects of interest were smaller than R^2^ = 6%, so our study may be underpowered. This problem could be addressed in future studies by increasing the sample size. Second, the pregnant women were recruited through an online advertisement, whereas the non-pregnant women were recruited directly at the university, resulting in the non-pregnant women being younger than the pregnant group. In addition, according to our analysis, the age of the participants was not a predictor of pain sensitivity indicators. However, future studies focusing on a more in-depth analysis of the effects of demographic characteristics on pain sensitivity should use uniform methods to recruit participants to include more heterogeneous groups in terms of age and other sociodemographic factors. Third, short measures of personality traits, such as TIPI, might not be specific enough and often suffer from low internal consistency. Therefore, future studies should employ measures of personality that have better reliability and validity, such as NEO-FFI or HEXACO [58,59] Fourth, one of the weaknesses of the study was that our classification of participants’ physical activity levels as low, moderate, or high was not based on objective methods. Although the IPAQ is a standardized and commonly used tool to assess physical activity levels in different populations, the use of objective devices such as accelerometers could potentially alter the observations and provide better insight into the relationship between physical activity and pain tolerance. Fifth, the study presented here is a cross-sectional study. Although differences between pregnant and non-pregnant women and associations between pain sensitivity and personality characteristics were observed, the data are limited to a single time point. Experimental studies with lifestyle interventions, including promoting optimism, managing fears, and anxieties, and performing physical activities of varying intensity [60], as well as longitudinal studies conducted throughout pregnancy, would certainly provide more information on the extent to which pregnancy itself and lifestyle factors influence pain threshold, pain tolerance, and subjective ratings of pain. Finally, the 21st century brought climate change, pandemics, and wars. The COVID-19 pandemic caused by the SARS-CoV-2 virus proved to be the largest pandemic in this century. We collected the data in the first half of 2021, when the COVID-19 situation in Poland could have been stressful for the community: There were many restrictions on social contacts, health services, and joint entertainment. It is very likely that this situation could affect the individuals’ well-being. Pregnant women are a particular group in which strong negative emotions can lead to pregnancy complications and affect the mother’s well-being, the course of pregnancy, and the child’s condition. Despite the above-mentioned limitations of our work, it may be important for policymakers to take action to promote a healthy lifestyle among pregnant women as well as support natural childbirth. A better understanding of the mechanisms regulating pain may translate into limiting the use of anesthetics during childbirth, thus reducing the side effects of surgical deliveries and increasing the self-esteem of health and well-being of both mother and offspring. The results of studies on pain perception in pregnant women provide valuable information for the medical team to optimize the birth process. Therefore, it seems justified to explore the physiological and psychological mechanisms that modulate pain not only in pregnant women but in the whole population.

## 5. Conclusions

In summary, our study showed that pregnant women had a significantly lower pain tolerance compared with non-pregnant women, contradicting the notion that late pregnancy is associated with increased antinociception due to enhanced release of β-endorphins. Analysis of predictors of pain sensitivity, such as level of COVID-19 anxiety, optimism, personality traits, and level of physical activity, which significantly modulate pain perception in a population of healthy individuals in pregnant women, revealed no correlation with pain tolerance. This fact proves that pain measurements in people under stress may affect the perception of pain stimuli applied experimentally from the application. Moreover, the data presented in the paper show that the differences in pain tolerance between the two groups cannot be due to age or life stage.

## Figures and Tables

**Table 1 ijerph-19-14151-t001:** Pain sensitivity indicators in pregnant and non-pregnant women.

Variables	Non-Pregnant Women n = 59		Pregnant Women n = 66		*t*	*df*	*p*	Cohen’s *d*
	*M* ± *SD*		*M* ± *SD*					
PPT dominant hand	89.63	±64.70	97.58	±57.96	−0.71	113.54	0.478	−0.129
PPT non-dominant hand	117.25	±93.27	109.72	±66.45	0.51	99.40	0.612	0.093
PTOL dominant hand	1159.23	±326.30	713.47	±259.36	8.30	106.47	**<0.001**	**1.512**
PTOL non-dominant hand	1083.88	±358.64	667.62	±237.97	7.46	94.87	**<0.001**	**1.368**
VAS dominant hand	5.97	±1.66	5.99	±1.68	−0.07	118.80	0.947	−0.012
VAS nondominant hand	6.37	±1.69	6.76	±1.53	−1.33	114.19	0.186	−0.242

Note: PPT = pain threshold, PTOL = pain tolerance, VAS = subjective assessment of pain. The data are presented as means and standard deviation. Values in bold represent statistically significant differences in the Welsh’s *t*-test at *p* < 0.05.

**Table 2 ijerph-19-14151-t002:** Psychological traits assessed in pregnant and non-pregnant women.

Variables	Non-Pregnant women n = 55		Pregnant Women n = 66		*t*	*df*	*p*	Cohen’s *d*
	*M* ± *SD*		*M* ± *SD*					
Optimism	15.53	±3.58	17.03	±3.85	−2.22	117.53	**0.028**	**−0.405**
Extraversion	11.98	±2.17	11.85	±2.12	0.34	114.06	0.735	0.062
Agreeableness	11.75	±1.94	11.32	±1.88	1.22	113.88	0.223	0.224
Conscientiousness	10.60	±2.42	11.68	±2.30	−2.51	112.77	**0.014**	**−0.459**
Emotional stability	8.75	±3.04	9.42	±3.19	−1.20	116.81	0.235	−0.218
Openness to experience	10.40	±1.80	10.30	±1.85	0.29	116.07	0.771	0.053
Fear of COVID-19	10.69	±3.30	13.09	±4.52	−3.37	117.08	**0.001**	**−0.607**

The data are presented as means and standard deviation. Values in bold represent statistically significant differences in the Welsh’s *t*-test at *p* < 0.05.

**Table 3 ijerph-19-14151-t003:** Correlations between pain sensitivity indicators and psychological traits in the total sample, in non-pregnant women.

Psychological Variables	Pain Threshold				Pain Tolerance				Subjective Pain Assessment			
	DH		NDH		DH		NDH		DH		NDH	
	ρ	*p*	ρ	*p*	ρ	*p*	ρ	*p*	ρ	*p*	ρ	*p*
Total sample (n = 121)												
Optimism	**−0.25**	**0.005**	**−0.25**	**0.006**	−0.17	0.058	−0.13	0.144	0.06	0.543	0.07	0.449
Extraversion	−0.13	0.160	**−0.18**	**0.044**	−0.03	0.731	0.00	0.970	0.12	0.189	**0.20**	**0.030**
Agreeableness	−0.13	0.157	0.06	0.551	0.03	0.775	0.10	0.257	−0.16	0.085	−0.14	0.134
Conscientiousness	−0.07	0.474	**−0.26**	**0.004**	−0.17	0.066	**−0.21**	**0.019**	−0.01	0.958	0.09	0.329
Emotional Stability	−0.10	0.264	−0.13	0.172	−0.14	0.125	−0.10	0.274	0.00	0.962	0.08	0.386
Openness to experience	−0.06	0.488	−0.13	0.149	0.09	0.322	0.06	0.489	−0.09	0.342	0.01	0.882
Fear of COVID-19	−0.03	0.717	0.02	0.860	−0.21	0.019	−0.15	0.113	0.12	0.196	−0.08	0.381
Non-pregnant women (n = 55)												
Optimism	−0.14	0.326	−0.10	0.480	−0.01	0.955	−0.02	0.911	−0.03	0.844	0.02	0.860
Extraversion	−0.17	0.206	**−0.28**	**0.042**	0.02	0.896	−0.11	0.436	0.01	0.940	0.12	0.399
Agreeableness	−0.21	0.128	0.07	0.633	0.10	0.479	**0.30**	**0.027**	−0.16	0.239	−0.15	0.279
Conscientiousness	−0.14	0.310	**−0.39**	**0.003**	0.00	0.982	−0.16	0.243	0.10	0.478	0.23	0.093
Emotional Stability	−0.03	0.853	−0.07	0.629	−0.02	0.906	−0.01	0.930	−0.05	0.718	−0.08	0.566
Openness to experience	−0.07	0.619	0.01	0.967	0.04	0.784	−0.04	0.763	−0.15	0.274	−0.01	0.932
Fear of COVID-19	0.05	0.736	0.04	0.786	−0.04	0.761	−0.06	0.669	0.19	0.165	−0.03	0.812
Pregnant women (n = 66)												
Optimism	**−0.33**	**0.006**	**−0.27**	**0.026**	−0.08	0.545	−0.04	0.731	**0.26**	**0.035**	**0.24**	**0.048**
Extraversion	−0.03	0.824	−0.11	0.374	−0.14	0.247	−0.03	0.841	**0.30**	**0.014**	**0.32**	**0.009**
Agreeableness	−0.02	0.849	0.08	0.504	−0.17	0.186	−0.20	0.118	−0.19	0.13	−0.17	0.171
Conscientiousness	−0.02	0.855	−0.03	0.839	−0.03	0.820	−0.01	0.967	0.02	0.881	−0.01	0.970
Emotional Stability	−0.05	0.722	−0.15	0.223	−0.09	0.455	−0.08	0.543	0.11	0.378	**0.26**	**0.034**
Openness to experience	−0.09	0.461	−0.16	0.194	0.16	0.197	0.17	0.178	0.02	0.896	0.10	0.412
Fear of COVID-19	−0.21	0.088	−0.10	0.412	−0.04	0.780	0.06	0.652	0.16	0.21	−0.09	0.471

Note: DH = dominant hand, NDH = non-dominant hand. Values in bold represent statistically significant correlations at *p* < 0.05.

**Table 4 ijerph-19-14151-t004:** General linear model for the predictors of pain sensitivity indicators.

Variable	Dominant Hand				Non-Dominant Hand			
	β	se	*t* (110)	*p*	β	se	*t* (110)	*p*
Dependent variable: Pain threshold								
Pregnant vs. non-pregnant	−0.23	0.21	−1.07	0.285	−0.34	0.21	−1.61	0.110
Physical activity	−0.14	0.12	−1.19	0.237	−0.13	0.12	−1.12	0.264
Age	**0.38**	**0.18**	**2.13**	**0.035**	**0.42**	**0.18**	**2.35**	**0.021**
Optimism	**−0.30**	**0.11**	**−2.73**	**0.007**	−0.16	0.11	−1.44	0.154
Extraversion	−0.01	0.10	−0.05	0.957	−0.12	0.10	−1.16	0.251
Agreeableness	−0.07	0.09	−0.75	0.455	0.17	0.09	1.78	0.079
Conscientiousness	−0.09	0.10	−0.88	0.381	**−0.22**	**0.10**	**−2.30**	**0.023**
Emotional stability	0.12	0.11	1.10	0.274	0.01	0.11	0.08	0.938
Openness to experience	−0.09	0.10	−0.96	0.341	−0.05	0.10	−0.56	0.579
Fear of COVID-19	−0.18	0.11	−1.74	0.085	−0.12	0.10	−1.13	0.260
Dependent variable: Pain tolerance								
Pregnant vs. non-pregnant	**−0.68**	**0.18**	**−3.78**	**<0.001**	**−0.66**	**0.19**	**−3.56**	**<0.001**
Physical activity	−0.16	0.10	−1.56	0.122	−0.15	0.11	−1.43	0.157
Age	−0.01	0.15	−0.06	0.953	0.04	0.16	0.28	0.780
Optimism	−0.02	0.09	−0.25	0.804	−0.02	0.10	−0.17	0.865
Extraversion	−0.06	0.09	−0.71	0.483	−0.06	0.09	−0.67	0.503
Agreeableness	−0.02	0.08	−0.19	0.853	0.10	0.08	1.16	0.248
Conscientiousness	0.00	0.08	−0.02	0.984	−0.08	0.09	−0.97	0.334
Emotional stability	−0.04	0.10	−0.38	0.706	−0.02	0.10	−0.18	0.854
Openness to experience	0.10	0.08	1.17	0.245	0.06	0.09	0.70	0.483
Fear of COVID-19	−0.02	0.09	−0.26	0.796	0.01	0.09	0.07	0.947
Dependent variable: Subjective pain assessment								
Pregnant vs. non-pregnant	−0.06	0.21	−0.29	0.770	0.08	0.21	0.39	0.696
Physical activity	0.10	0.12	0.84	0.400	0.05	0.12	0.43	0.668
Age	0.01	0.18	0.08	0.939	0.03	0.18	0.19	0.851
Optimism	0.15	0.11	1.32	0.191	0.10	0.11	0.92	0.360
Extraversion	0.17	0.10	1.65	0.103	**0.21**	**0.10**	**2.00**	**0.048**
Agreeableness	**−0.21**	**0.10**	**−2.19**	**0.030**	**−0.23**	**0.10**	**−2.45**	**0.016**
Conscientiousness	0.06	0.10	0.63	0.530	0.05	0.10	0.54	0.593
Emotional stability	0.02	0.11	0.15	0.882	0.01	0.12	0.11	0.912
Openness to experience	−0.06	0.10	−0.58	0.561	−0.03	0.10	−0.26	0.795
Fear of COVID-19	0.18	0.11	1.68	0.096	−0.07	0.11	−0.62	0.537

Note: DH = dominant hand, NDH = non-dominant hand. Values in bold represent statistically significant predictors at *p* < 0.05.

## Data Availability

The data can be made available from the corresponding author (K.L.) upon reasonable request.

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
