# Peer review of "Individual Characteristics and Pain Sensitivity during Pregnancy—A Cross-Sectional Study in Pregnant and Non-Pregnant Women"

_ijerph, 2022, doi:10.3390/ijerph192114151_

Round 1

Reviewer 1 Report

Introduction:

 -The authors point out the increased incidence of depressive disorders and fear in pregnant women due to the covid-19 pandemic, they bring to the fore the topic of subjectively experienced pain, which is not often investigated by pregnant women, and point to the connection with personality traits, which I rate as very positive and beneficial. They also highlight the benefits of optimism and humor and their impact on the mental health of mothers, which has decreased due to the pandemic and caused an increase in fear among pregnant women.

- All psychological constructs are terminologically explained and comprehensible, and the content is clearly processed into individual paragraphs

Materials and Methods

-The materials and methods part is processed in detail, it contains all important information, including a detailed description of the research tools, the research process, and the ethical requirements of data collection

Results:

- The authors performed a comparative, correlational, and regression analysis, and the results are written down in detail and clearly. They contain explanations and descriptions.

Discussion:

-The discussion is engaging, provides a comprehensive overview of the findings, and supports them with over 50 relevant research studies

I congratulate the authors for a job well done, I read the study carefully and have nothing to criticize the authors. That is why I recommend the editors accept the study also in this form, it certainly has the potential to bring valuable information on this issue.

Author Response

Dear Reviewer,

We thank you for your appreciation of our work. We have corrected the conclusion as suggested

Authors

Reviewer 2 Report

I realize that great work and time have been devoted to this paper. It has a lot of strengths, but I think that some changes should be recommended. 

Title: the title is too long. Please, try to change it to better inform the readers about the relationships between the variables that you test and also inform them about the quality of your sample, but reduce its length.

Abstract:

Also, the abstract includes statistical values that are not necessary.

Introduction

The literature revision has some references that are too old. It is ok having 56 references, but only a small part of them are from the last 5 years. Please, update your literature review.

Methodology

The Instruments or Questionnaires section is a bit incomplete. If you can, please inform me about previous studies where the same instrument has been used and the reliability obtained in that research. Moreover, some criticism has been expressed on the TIPI scale. Please, justify your decision of using it or discuss the critiques in the limitations section.

Results

I am not absolutely sure of understanding why you compare the participants’ results in the dominant and non-dominant hands. Please, can you better explain it?

Discussion:

First of all, try to better adjust your conclusions to the findings. Or to say in other words, please try to justify more clearly the connection between your conclusions and your findings.

Finally, your paper has a lot of relevant implications for society and policymakers, but you need to elaborate more on this topic.

Author Response

Dear Reviewer,

We would like to thank you for your careful reading of our manuscript and insightful comments, which helped us to improve our work significantly. We have revised the manuscript to address the reviewer's comments.

Authors

I realize that great work and time have been devoted to this paper. It has a lot of strengths, but I think that some changes should be recommended.

Title: the title is too long. Please, try to change it to better inform the readers about the relationships between the variables that you test and also inform them about the quality of your sample, but reduce its length. 

Authors:     We changed the title as suggested.

Abstract:

Also, the abstract includes statistical values that are not necessary. 

Authors:   Thank you for your attention. We improved according to your suggestion

Introduction

The literature revision has some references that are too old. It is ok having 56 references, but only a small part of them are from the last 5 years. Please, update your literature review.  

Authors:   Thank you for your attention. We have tried to introduce more recent references.

Methodology

The Instruments or Questionnaires section is a bit incomplete. If you can, please inform me about previous studies where the same instrument has been used and the reliability obtained in that research. Moreover, some criticism has been expressed on the TIPI scale. Please, justify your decision of using it or discuss the critiques in the limitations section.

Authors: We thank you for these comments. We have made appropriate descriptions in the manuscript

Results

I am not absolutely sure of understanding why you compare the participants’ results in the dominant and non-dominant hands. Please, can you better explain it? 

Authors:  Human mobility is based on different functions of the limbs. The dominant limb differs morphologically, physiologically, and functionally from the nondominant limb because of its more frequent involvement in many motor activities. The significantly more frequent performance of motor tasks by the dominant limb also directly or indirectly influences the perception of mechanical and thermal pain stimuli, hence we have included this aspect in our research.

 Discussion:

First of all, try to better adjust your conclusions to the findings. Or to say in other words, please try to justify more clearly the connection between your conclusions and your findings.

Finally, your paper has a lot of relevant implications for society and policymakers, but you need to elaborate more on this topic. 

Authors:  We have corrected the parts of the manuscript as suggested.